# Mycoviral Diversity of *Fusarium oxysporum* f. sp. *niveum* in Three Major Watermelon-Production Areas in China

**DOI:** 10.3390/microorganisms13040906

**Published:** 2025-04-14

**Authors:** Jiawang Yang, Yajiao Wang, Zihao Li, Sen Han, Bo Li, Yuxing Wu

**Affiliations:** 1Institute of Plant Protection, Hebei Academy of Agricultural and Forestry Sciences, Baoding 071000, China; 16630393800@163.com (J.Y.); yajiaowang515@163.com (Y.W.); hansen19920603@163.com (S.H.); 2State Key Laboratory of North China Crop Improvement and Regulation, College of Plant Protection, Hebei Agricultural University, Baoding 071000, China; hbnylizihao@163.com

**Keywords:** *Fusarium* wilt of watermelon, *Fusarium oxysporum* f. sp. *niveum*, mycovirus, diversity

## Abstract

Watermelon is one of the most important fruits in China, accounting for more than 70% of the world’s total output. *Fusarium* wilt of watermelon is the most common and serious disease in the cultivation of watermelon. It is mainly caused by *Fusarium oxysporum* f. sp. *niveum* (FoN), which has caused serious damage to the watermelon-planting industry. Some mycoviruses can reduce the pathogenicity of host pathogens and have the potential for biocontrol, so their application potential in the biological control of plant fungal diseases has attracted much attention. In this study, high-throughput sequencing was performed on 150 FoN strains isolated from three major watermelon-production areas (northern semi-arid area, northwestern arid area, and southern humid area) to detect the diversity of mycoviruses and to uncover new mycoviruses. The analysis identified 25 partial or complete genome segments representing eight previously undescribed mycoviruses. The existence of six mycoviruses was verified via RT-PCR. The southern humid area had the highest diversity of mycoviruses, with 15 species identified. Among these, 40% are dsRNA viruses and 33.3% belong to the family Chrysoviridae, representing the predominant viral type and family. In the northern semi-arid area, a total of 12 viral species were identified, among these 41.7% were +ssRNA viruses and 25% belonged to the family Mymonaviridae, constituting the main viral types and family. The northwestern arid area showed relatively low viral diversity, only containing three species. Two of these were +ssRNA viruses classified under the Mitoviridae and Potyviridae families. Notably, only one virus, *Fusarium oxysporum* f. sp. *niveum* Potyvirus 1 (FoNPTV1), was shared across all three areas. These findings reveal significant regional differences in the mycoviral species composition and distribution, highlighting the complex interactions between mycoviruses and FoN in different environments. By uncovering new mycoviruses associated with FoN, this study provides valuable resources for the potential biocontrol of *Fusarium* wilt in watermelon, contributing to sustainable disease management and improving the quality and safety of watermelon production in China.

## 1. Introduction

Watermelon is an important economic crop in China. According to the National Bureau of Statistics, between 2018 and 2022, China’s watermelon-cultivation area reached 1.5 million hectares, and the average yield of watermelons was about 62.35 million tons (https://data.stats.gov.cn/easyquery.htm?cn=C01 (accessed on 10 February 2025)). China is the largest producer of watermelons in the world [1]. Based on the climate, geographical conditions, and cultivation practices, China’s watermelon-production areas can be divided into three major regions: the northern semi-arid area, northwestern arid area, and southern humid area. The northern semi-arid area experiences cold winters and warm, dry summers, with low annual rainfall, but has fertile soil, resulting in high watermelon yields. The northwestern arid area is characterized by dry conditions with little rainfall and large temperature differences between day and night, leading to high-quality watermelons with high sugar contents. The southern humid area has a humid and rainy climate, supporting multi-season cultivation, with watermelons coming to market early and having a long supply period.

Watermelon is threatened by various diseases throughout its growth period [2]. Among them, *Fusarium* wilt of watermelon, caused by *Fusarium oxysporum* f. sp. *niveum* (FoN), is present in watermelon-production areas around the world [3]. FoN invades the plant through root hairs or wounds in the roots. It multiplies and spreads within the vascular vessels, producing toxins that damage cells and block the vessels, disrupting water transport. This leads to wilting or death of the plant, causing significant losses in watermelon production [4]. In recent years, *Fusarium* wilt of watermelon has become increasingly severe in China due to continuous cropping obstacles, climate change, and the lack of disease-resistant varieties. Conventional control strategies for FoN primarily rely on resistant cultivars and fungicide applications, yet these approaches face limitations in long-term efficacy and environmental safety [5,6]. Although breeding resistant varieties is considered one of the most effective strategies for controlling *Fusarium* wilt of watermelon, the development of commercially viable and highly resistant cultivars remains limited, primarily due to challenges, such as prolonged breeding cycles and inefficient selection processes. Chemical control methods for *Fusarium* wilt of watermelon include the use of prothioconazole and thiophanate-methyl. [6]. But it has been difficult to solve defects, and the long-term use of chemical agents can lead to drug resistance, environmental pollution, and harm to non-target organisms and results in pesticide residues. Therefore, people are increasingly interested in exploring effective and environmentally friendly alternative methods for controlling the disease. Biological control has the advantages of low pollution, high specificity, and environmental friendliness and has broad application prospects [7]. Therefore, environmentally friendly biological control, including the use of hypovirulence-related mycoviruses for control, is considered a feasible alternative method [8].

Mycoviruses are viruses that infect fungi and can replicate within fungal cells [9]. Studies have shown that most fungal viruses exhibit asymptomatic infection of the host, but some fungi may exhibit phenotypic abnormalities, such as an abnormal colony morphology, reduced spore production, slower mycelial growth, decreased pathogenicity, or increased pathogenicity after being infected [10,11]. These abnormalities can reduce the host’s virulence, thus providing an environmentally friendly and sustainable means of disease control. Attenuated mycoviruses have been reported in a variety of fungal species, including *Fusarium graminearum* [12], *Colletotrichum* spp. [13], and *Penicillium digitatum* [14]. In practical applications, certain mycoviruses known to induce hypovirulence in fungi have been developed as biocontrol agents against plant fungal diseases. For instance, *Cryphonectria* hypovirus 1 (CHV1), which infects the chestnut blight fungus *Cryphonectria parasitica*, has been successfully used for the biological control of chestnut blight in Europe [15]. Similarly, *Pestalotiopsis theae* chrysovirus-1 (PtCV1) [16] in the tea gray blight pathogen and *Leptosphaeria biglobosa* quadrivirus 1 (LbQV1) [17] in the blackleg pathogen of oilseed rape have been found to enhance plant resistance to their respective fungal hosts. Additionally, it has been reported that spraying *Brassica napus* at the early flowering stage with a strain of *Sclerotinia sclerotiorum* hypovirulence-associated DNA virus 1 (SsHADV-1)-infected DT-8 can reduce the severity of stem rot by 67.6% and increase the yield by 14.9% [18]. The large-scale application of existing mycoviral formulations is limited by the low transmission efficiency, narrow host range, and insufficient stability, making them less effective in complex field disease systems. Therefore, it is crucial to identify novel mycoviruses with high transmission efficiency, broad-spectrum antifungal activity, and genetic stability to enhance biocontrol effectiveness.

Despite growing interest in mycoviruses as potential biocontrol agents, their areal distribution, diversity, and ecological roles in FoN remain largely unexplored. In this study, we employed high-throughput sequencing (HTS) to systematically analyze the virome of 150 FoN strains isolated from the three major watermelon-producing regions in China. Our objectives were threefold: (1) to characterize the diversity and composition of mycoviruses associated with FoN populations, (2) to identify novel mycoviruses with potential hypovirulence-inducing properties, and (3) to elucidate areal patterns in mycovirus distribution and their implications for biocontrol. This study represents the first large-scale areal analysis of FoN mycoviruses in China, contributing to sustainable disease management and improved watermelon production.

## 2. Materials and Methods

### 2.1. Fungus Isolation and Purification

Watermelon samples infected with FoN were collected from three key watermelon-producing areas in China: Hebei Province (36°01′ N–42°37′ N, 113°04′ E–119°53′ E) in the northern semi-arid area, Fujian Province (23°30′ N–28°22′ N, 115°50′ E–124°07′ E) in the southern humid area, and Xinjiang Uygur Autonomous Region (34°22′ N–49°15′ N, 73°40′ E–96°18′ E) in the northwestern arid area. The northern semi-arid area is characterized by a mid-latitude steppe climate, with an annual average temperature of 16.38 °C, which is 1.76% higher than the national average in China. Its annual precipitation is typically around 20.33 mm, and the humidity level is 47.18%. (https://weatherandclimate.com/ (accessed on 16 July 2024)). In the northern semi-arid area, watermelon cultivation is primarily based on a one-crop-per-year system. The southern humid area has a humid subtropical climate with no dry season. Its annual average temperature is 19.79 °C, 5.17% higher than the national average in China. The annual precipitation is typically around 87.27 mm, and the humidity level is 82.02%. In the southern humid area, watermelon cultivation is primarily based on a two-crop-per-year system. In some areas, through the application of facility cultivation and early-maturing varieties, an efficient three-crop-per-year planting model can be achieved. The northwestern arid area is classified as having a mid-latitude desert climate. Its annual average temperature is 3.13 °C, which is 11.49% lower than the national average in China. The annual precipitation is typically around 14.75 mm, and the humidity level is 45.83%. The northwestern arid area has a relatively cold climate and a short frost-free period, so watermelon cultivation is typically based on a one-crop-per-year system. We collected 200, 223 and 178 watermelon *Fusarium* wilt samples from the northern semi-arid area, northwestern arid area, and southern humid area, respectively, with a total of 601 samples. Through pathogen isolation, purification, morphological identification, and molecular characterization, we isolated 140, 162, and 113 strains of 415 FoN from the northern semi-arid area, the southern humid area, and the northwestern arid area, respectively, and purified and identified them using established methods [19].

Fifty strains were randomly selected from each producing area for biological characterization and sequencing, and the selected strains were cultured on PDA at 25 °C. The colony diameter was measured daily for 5 days using the cross method, with three replicates per day. Average values were calculated, and the mycelial growth rate was determined. After 5 days of culture, the morphology of the cultured strain was observed and photographed. The selected strains were inoculated respectively into 100 mL of Potato Dextrose Broth (PDB). The cultures were incubated in a shaking incubator at 25 °C and 180 rpm for 3 days. Post-cultivation, mycelia were removed via filtration through a sterilized filter cloth. The spore yield was estimated using a five-point sampling method combined with hemocytometer counting. To ensure reliability, all experiments were conducted in triplicate for each strain.

### 2.2. Extraction of Total RNA

The selected strains were cultured respectively on cellophane-overlaid PDA media (PDA-CF) at 25 °C in complete darkness for 5 days. After incubation, the mycelia of 50 strains from the same area were mixed together, ground into a fine powder using a mortar and pestle with liquid nitrogen, and subjected to total RNA extraction using the TransZolUpPlusRNA kit (TransGenBiotech, Beijing, China) following the manufacturer’s protocol. The quality and quantity of the extracted RNA were evaluated using two complementary methods: spectrophotometric analysis with a Nanodrop 2000 system (Thermo Scientific, Waltham, MA, USA) and electrophoretic separation on a 1.0% agarose gel. To ensure consistency in downstream applications, RNA samples were normalized to a uniform concentration of 200 ng/μL.

### 2.3. Library Preparation and Sequencing

RNA purification, reverse transcription, library preparation, and high-throughput sequencing were conducted by Shanghai Majorbio Bio-pharm Biotechnology Co., Ltd. (MajorBio, Shanghai, China). For each sample, 1 μg of total RNA was processed to construct strand-specific sequencing libraries after ribosomal RNA (rRNA) depletion. Libraries were prepared using the Illumina Stranded Total RNA Prep, with ligation using the Ribo-Zero Plus kit, strictly adhering to the manufacturer’s protocols. The pooled libraries were then subjected to paired-end sequencing on an Illumina NovaSeq X Plus platform (Illumina, San Diego, CA, USA). Raw sequencing data often contain adapter sequences, low-quality bases, uncertain bases (denoted as N), and short sequences, which can compromise the accuracy of downstream analyses. To address this, we first used fastp (v0.19.6, available at https://github.com/OpenGene/fastp (accessed on 1 February 2024)) [20] to perform quality control. This involved trimming adapter sequences from the 3′ and 5′ ends of the reads. Reads shorter than 50 nt after trimming, with an average quality score below 20, or containing N bases were discarded, ensuring that only high-quality paired-end reads were retained. Next, we performed de novo assembly on the cleaned reads using MEGAHIT (v1.3.0, Dinghua Li, Shenzhen, China), generating contigs for each sample. Contigs with a length of ≥1 kb, and those identified as circular were selected for further analysis.

### 2.4. Virus-Related Sequence Assembly and Annotation

For viral contig identification, assembled sequences were screened against the NCBI nr protein database through DIAMOND BLASTX (v0.9.10, Benjamin Buchfink, Tübingen, Baden-Württemberg, Germany) [21] with a stringent e-value cutoff (1 × 10^−5^) to balance sensitivity and specificity. Contigs aligned to identical reference sequences were consolidated into extended contigs using DNAMAN (v8.0, Lynnon Biosoft, Vaudreuil, QC, Canada). Final sequence annotation, ORF prediction, and amino acid analyses were performed using DNAMAN. The raw sequencing data from the metavirome libraries have been deposited in the NCBI Sequence Read Archive (SRA) database under BioProject accession number PRJNA1158555.

### 2.5. Phylogenetic Analysis of Mycoviruses

To analyze the evolutionary relationships of the newly identified viruses, the RNA-dependent RNA polymerase protein (RdRp) or Coat Protein (CP) sequences encoded by the putative mycoviruses were compared using BLASTp against the NCBI-nr database. The core conserved domains of RdRp were aligned with MAFFT (v7.427, Kazutaka Katoh, Mishima, Japan) using the E-INS-i algorithm. The alignments were then refined with trimAl (v1.4, Salvador Capella-Gutiérrez, Madrid, Spain) to remove low-quality regions, applying the automated heuristic method based on similarity statistics.

The phylogenetic tree was constructed using MEGA7 software (Molecular Evolutionary Genetics Analysis v7.0, Sudhir Kumar, Tempe, AZ, USA). First, the target sequences were aligned using the ClustalW algorithm to ensure accurate sequence alignment. After alignment, the Neighbor-Joining (NJ) method in MEGA7 was employed to construct the phylogenetic tree. The NJ tree was built based on the tp-distance model, which is suitable for calculating evolutionary distances for nucleotide sequences. Bootstrap testing with 1000 replicates was performed to assess the node support of the phylogenetic tree, and nodes with bootstrap values greater than 70% were considered statistically well-supported. The final phylogenetic tree was visualized and edited using MEGA7.

### 2.6. Putative Mycovirus Sequence Confirmation

Total RNA of the 150 selected strains was extracted separately with TransZol Up (TransGen Biotech, Beijing, China) and quantified using a NanoDrop spectrophotometer (Thermo Scientific, Waltham, MA, USA). cDNA synthesis was performed using the Fastking gDNA Dispelling RT SuperMix kit (TIANGEN, Beijing, China) according to the manufacturer’s protocol. Two viruses from each of the three main nucleic acid types (dsRNA, +ssRNA, and −ssRNA) were detected via RT-PCR. Specific primers were designed with prime 5 software (Appendix A). PCR amplification was conducted with the M5 HiPer plus Taq HiFi PCR mix (2×, with blue dye) (Mei5bio, Beijing, China). The PCR conditions included an initial denaturation at 95 °C for 5 min, followed by 35 cycles of 95 °C for 45 s, 58 °C for 45 s, and 72 °C for 45 s, with a final extension at 72 °C for 5 min. RT-PCR products were separated through 1% agarose gel electrophoresis and visualized using a gel imaging system. The presence of the virus and the virus-carrying strains was verified through RT-PCR, and whether the area where the virus-carrying strain was collected matched the area where the virus was detected in the sequencing results was analyzed.

## 3. Results

### 3.1. Collection and Analysis of FoN from Three Major Watermelon-Production Areas

From the three major production areas (northern semi-arid area, northwestern arid area, and southern humid area), 50 strains were randomly selected for biological characterization and mycovirus diversity analysis. The results revealed significant differences in their biological characteristics, with 18% (27/150) exhibiting phenotypic abnormalities, including slow growth (6 strains), reduced sporulation (11 strains) (Appendix A), abnormal pigment production (5 strains), and abnormal colony morphology (5 strains) (Appendix A). Among these, 16% (8 strains) of the abnormal phenotypes were from the northern semi-arid area, 30% (15 strains) from the southern humid area, and 8% (4 strains) from the northwestern arid area.

### 3.2. Diversity and Abundances of FoN Mycoviruses in Three Areas

The total RNA of the strains was subjected to high-throughput sequencing, and the results were compared with a viral database using BLASTX for homologous alignment to identify mycoviruses in FoN. From these three areas, we identified a total of 25 mycoviruses, among which 8 had nucleic acid sequences of the RdRp or CP with less than 70% and 50% similarity, respectively, to known viruses, and these were thus classified as novel viruses according to The International Committee on Taxonomy of Viruses (ICTV) criteria (Appendix A). Of the 25 viruses, eight (32%) were positive-sense single-stranded RNA viruses (+ssRNA), six (24%) were double-stranded RNA viruses (dsRNA), four (16%) were negative-sense single-stranded RNA viruses (−ssRNA), three (12%) were retroviruses, and the remaining four consisted of two DNA viruses and two unclassified viruses each accounting for 8% (Figure 1a).

There were significant differences in the types and categories of mycoviruses in FoN among the three major watermelon-producing areas. In the northern semi-arid area, a total of 12 viruses were detected, including 5 novel viruses: FoNMTV1, FoNMTV2, FoNMV2, FoNNAV1, and FoNMGV1. In the southern humid area, 15 viruses, including three novel ones (FoNCHV2, FoNCHV4, FoNMLV2), were detected. Three viruses were detected in the northwestern arid area, none of which were novel (Figure 1b). Two mycoviruses, FoNPV1 and FoNHV1, were detected in both the northern and southern areas but not in the northwestern area. One mycovirus, FoNNLV1, was detected in both the northwestern and southern areas but not in the northern area. One mycovirus, FoNPTV1, was detected in all three areas (Figure 1c).

The southern humid area had six dsRNA viruses, four +ssRNA viruses, two DNA viruses, one retrovirus, and two unclassified viruses (Figure 1d). Among these, there were five mycoviruses belonging to the Chrysoviridae family, and one each from the Parvoviridae, Narnaviridae, Picobirnaviridae, Mitoviridae, Poxviridae, Potyviridae, Retroviridae, and Hadakaviridae families, with two remaining unclassified (Figure 1e). This indicates that the majority of mycoviruses in the southern humid area are dsRNA viruses, with the Chrysoviridae family being the most prevalent. The northern humid area had one dsRNA virus, five +ssRNA viruses, four −ssRNA viruses, and two retroviruses. Among them, there were three species belonging to the Mymonaviridae family, two species belonging to the Mitoviridae family, and one each from the Hadakaviridae, Narnaviridae, Picobirnaviridae, Caulimoviridae, Retroviridae, and Potyviridae families, with one remaining unclassified. This shows that the majority of mycoviruses in the northern semi-arid area are +ssRNA viruses, with a significant presence from the Mymonaviridae families. The northwestern arid area had two +ssRNA viruses and one unclassified virus, with the two +ssRNA viruses belonging to the Mitoviridae and Potyviridae families, indicating that the mycoviruses detected in the northwestern arid area are predominantly +ssRNA viruses. Overall, the southern humid area had the highest species of mycoviruses with 15 species, mainly dsRNA viruses, accounting for 40%. The northern semi-arid area followed with 12 species, primarily +ssRNA viruses, making up 41.7%. In the northwestern arid area, the species of viruses was more limited, with only three types, two of which were +ssRNA viruses, accounting for 66.7% (Figure 1f).

### 3.3. Double-Stranded RNA Mycoviruses

We have detected a total of six dsRNA mycoviruses, which belong to two families, Chrysoviridae and Picobirnaviridae. Among these, the genome sequencing lengths of four viruses exceed 3500 bp. They have been designated as *Fusarium oxysporum* f. sp. *niveum* chrysovirus 1 (FoNCHV1), FoNCHV2, FoNCHV3, and *Fusarium oxysporum* f. sp. *niveum* tick virus 1 (FoNTV1). The genome of FoNCHV1 was found to consist of four dsRNA segments, each being monocistronic, named dsRNA1, dsRNA2, dsRNA3, and dsRNA4, respectively. Specifically, dsRNA1 encodes the RdRp, dsRNA2 encodes the coat proteins, while dsRNA3 and dsRNA4 encode two proteins of unknown function (Figure 2a). The RdRp protein of FoNCHV1 shows the highest similarity, at 99.5%, with that of *Fusarium sachari* chrysovirus 1 (QIQ28417.1) [22]. The genome sequencing length of FoNCHV3 is 4378 bp, containing two non-overlapping ORFs encoding an RNA polymerase and a protein of unknown function, respectively (Figure 2a). The RdRp sequence of FoNCHV3 shows the highest similarity, at 93.9%, with that of *Fusarium oxysporum* chrysovirus 1 (YP_009665200.1). The genome sequencing length of FoNTV1 is 3530 bp, containing only one ORF that encodes RdRp (Figure 2a). The RdRp protein of FoNTV1 has an 85.4% homology with that of Nanning Chrys tick virus 1 (UYL95303.1).

The genome sequencing length of FoNCHV2 is 8760 bp, comprising two segments of 4418 bp and 4342 bp, each carrying an open reading frame (ORF) that encodes a CP and a protein of unknown function, respectively (Figure 2a). The CP of FoNCHV2 has the highest similarity of 46.9% with that of *Ilyonectria pseudodestructans* chrysovirus 1 (UVD33181.1). The CP sequence of FoNCHV2 was used to construct an NJ phylogenetic tree with the CP sequences of 12 viruses from two genera (*Alphachrysovirus* and *Betachrysovirus*) within the family Chrysoviridae and four viruses from the family Totiviridae. The NJ tree revealed that FoNCHV2 clusters with *Ilyonectria pseudodestructans* chrysovirus 1 from the family Chrysoviridae and exhibits a closer phylogenetic relationship to viruses within the genus *Alphachrysovirus* (Figure 2b). According to the classification criteria of the ICTV (where a CP similarity of less than 53% indicates a new species), combined with the results of the phylogenetic tree, FoNCHV2 is identified as a new member belong to the genus *Alphachrysovirus*.

### 3.4. Novel Positive-Sense Single-Stranded RNA Viruses

We have identified eight +ssRNA mycoviruses classified into four viral families: Mitoviridae, Hadakaviridae, Potyviridae, and Narnaviridae. Notably, three of these viruses, *Fusarium oxysporum* f. sp. *niveum* Hadakavirus 1 (FoNHV1), *Fusarium oxysporum* f. sp. *niveum* mitovirus 1 (FoNMTV1), and *Fusarium oxysporum* f. sp. *niveum* mito-like virus 1 (FoNMLV1), exhibited genomic sequences exceeding 2000 nucleotides (nt) in length. FoNHV1 was found to have a genome composed of ten dsRNA segments, each of which is monocistronic and contains a single open reading frame (ORF). The dsRNA1 segment is 2544 nt in size and encodes RdRp, while the dsRNA3 segment encodes a methyltransferase (MTR) (Figure 3a). The RdRp encoded by FoNHV1 shares 97% homology with that of Hadaka virus 1 (YP_010840281.1). FoNMLV1 has a genome sequencing length of 2256 nt, containing five non-overlapping ORFs of 456 nt, 375 nt, 264 nt, 253 nt, and 138 nt (Figure 3a), respectively, arranged linearly within the viral genome. The largest ORF encodes the RdRp. The RdRp sequence of FoNMLV1 shows the highest similarity (70.5%) to that of the known virus Henan mito-like virus 31 (UPW42176.1).

FoNMTV1 has a genome sequencing length of 2375 nt, containing three non-overlapping ORFs of 621 nt, 447 nt, and 201 nt (Figure 3a), respectively, arranged linearly within the viral genome. The largest ORF encodes the RdRp. The RdRp sequence of FoNMTV1 shows the highest similarity (64.1%) to that of the known virus *Botrytis cinerea* mitovirus 8 (QHJ68503.1). The RdRp sequence of FoNMTV1 was used to construct an NJ phylogenetic tree with the RdRp sequences of 10 viruses from three genera (*Mitovirus*, *Unuamitovirus*, and *Triamitovirus*) within the family Mitoviridae and two viruses from the family Chrysoviridae. The NJ tree revealed that FoNMTV1 clusters most closely with *Botrytis cinerea* mitovirus 8 from the genus *Mitovirus* within the family Mitoviridae, forming a distinct branch (Figure 3b). According to the classification criteria of the ICTV (where an RdRp sequence similarity of less than 70% indicates a new species of the genus *Mitovirus*, and combined with the results of the phylogenetic tree, FoNMTV1 is identified as a new member of the genus *Mitovirus*.

### 3.5. Novel Negative-Sense Single-Stranded RNA Viruses

We detected a total of four -ssRNA mycoviruses, three of which belong to the Mymonaviridae family. Their genome sequencing lengths all exceed 5000 nt, namely *Fusarium oxysporum* f. sp. *niveum* mymonavirus 1 (FoNMV1), FoNMV2, *Fusarium oxysporum* f. sp. *niveum* RNA virus 1 (FoNNAV1), and *Fusarium oxysporum* f. sp. *niveum* mononegavirus 1 (FoNMGV1). FoNMV1 has a sequencing length of 7039 nt, with its genome consisting of a single RNA segment containing two non-overlapping ORFs (Figure 4a), sized 5880 nt and 813 nt, encoding an RdRp and a protein of unknown function, respectively. The RdRp encoded by FoNMV1 shows the highest similarity (73.5%) to that of *Plasmopara viticola* lesion-associated mymonavirus 1 (YP_010798438.1).

FoNMV2 has a sequencing length of 6714 nt (Figure 4a), with its genome composed of a single RNA segment containing only one ORF of 1164 nt, which encodes the RdRp. The RdRp encoded by FoNMV2 shows the highest similarity (29.1%) to that of *Magnaporthe oryzae* mymonavirus 2 (QVU39969.1). FoNNAV1 has a sequencing length of 5119 nt (Figure 4a), with its genome consisting of a single RNA segment containing only one ORF of 1098 nt, which encodes the RdRp. The RdRp encoded by FoNNAV1 shows the highest similarity (52.8%) to that of Grapevine-associated negative single-stranded RNA virus 1 (QXN75409.1). Since the similarity between sequences FoNMV2 and FoNNAV1 and known sequences is very low, and there are few sequences similar to them, a phylogenetic tree was not constructed.

FoNMGV1 has a sequencing length of 6457 nt (Figure 4a), with its genome composed of a single RNA segment containing one large ORF of 5997 nt, which encodes the RdRp. The RdRp encoded by FoNMGV1 shows the highest similarity (46.2%) to that of *Plasmopara viticola* lesion-associated mononegaambi virus 8 (YP_010798440.1). The RdRp sequence of FoNMGV1 was used to construct an NJ tree with the RdRp sequences of 15 viruses from six genera (*Plasmopamonavirus*, *Penicillimonavirus*, *Lentimonavirus*, *Hubramonavirus*, *Botrytimonavirus*, and *Sclerotimonavirus*) within the family Mymonaviridae and three viruses from the family Megabirnaviridae. The NJ tree revealed that FoNMGV1 clusters most closely with members of the genus *Plasmopamonavirus* within the family Mymonaviridae, forming a distinct branch (Figure 4b). According to the classification criteria of the ICTV (where an RdRp sequence similarity of less than 70% indicates a new species), combined with the results of the phylogenetic tree, FoNMGV1 is identified as a new member of the genus *Plasmopamonavirus*.

### 3.6. Verification of Viral Genome Sequences via RT-PCR

To verify the existence of mycoviruses and the reliability of the viral sequences obtained through high-throughput sequencing, we randomly selected two viruses from each of the three main nucleic acid types (dsRNA, +ssRNA, and −ssRNA) for RT-PCR validation. The six mycoviruses selected were dsRNA viruses FoNTV1 and FoNCHV2, +ssRNA viruses FoNMTV1 and FoNMLV1, and −ssRNA viruses FoNNAV1 and FoNMV1. Two pairs of primers were designed for each virus, and the primers and target fragment sizes are listed in Appendix A. The RT-PCR validation results revealed the presence of FoNNAV1 and FoNMV1 in strain 64-1 from the northern semi-arid area, FoNMTV1 in strain 676-4, FoNTV1 and FoNCHV2 in strain 582 from the southern humid area, and FoNMLV1 in strain 472 from the northwestern arid area (Figure 5). The results indicate that the six viruses tested are indeed present, and their distribution across the area is consistent with the high-throughput sequencing results, confirming the reliability of the high-throughput sequencing data.

## 4. Discussion

In this study, we found that the diversity of mycoviruses in *Fusarium oxysporum* f. sp. *niveum* differ significantly in different areas. Out of the 25 detected mycoviruses, 15 are from the southern humid area and belong to the Chrysoviridae, Parvoviridae, Narnaviridae, Picobinaviridae, Mitoviridae, Poxviridae, Potyviridae, Retroviridae and Hadakaviridae families. Twelve species are from the northern semi-arid area, belonging to the Mymonaviridae, Hadakaviridae, Mitoviridae, Narnaviridae, Caulimoviridae, Picobinaviridae, Retroviridae, and Potyviridae families. The northwestern arid area yielded a comparatively limited diversity, with only three mycoviruses identified, which were classified within the Mitoviridae and Potyviridae families. There are significant differences in the number and types of viruses among the three major areas. We speculate that this is closely related to the geographical and climatic conditions of the three areas. The southern humid area is dominated by a tropical subtropical monsoon climate, with high temperatures and humidity, and precipitation of over 800 mm. Due to the favorable growing conditions, watermelons can be planted multiple times throughout the year, providing an extended survival environment for FoN, the host of mycoviruses [23,24]. The favorable growing conditions and extended survival environment have made this area have the richest variety of mycoviruses. The northwest area has a temperate continental dry climate, with dry conditions, low rainfall, large diurnal temperature variations, and significant annual temperature differences. The annual precipitation is generally low, with most places having an annual precipitation of less than 200 mm [25,26]. Due to climate limitations, watermelons in the northwest area are generally only planted once with the growing period from May to August. Compared to the southern humid area, this provides a shorter growth environment for the host of mycoviruses FoN, the host of mycoviruses. The unfavorable growing conditions and shorter growth environment result in a lower diversity of mycoviruses in this area.

Although most mycovirus infections do not have a significant impact on their fungal hosts, a few hypovirulence-related mycoviruses can cause a decrease in the growth rate, pathogenicity, and other phenotypes of their host fungi, which has the potential to control plant fungal diseases for biocontrol purposes. In this study, we preliminarily classified the collected strains based on their growth rate, sporulation quantity, and colony morphology, and found that 18% (27/150) of the strains had abnormal biological characteristics. According to previous reports, although many mycoviruses have been identified in the genus *Fusarium*, there are relatively few reported mycoviruses in *F. oxysporum*. For example, *Fusarium oxysporum* ourmia-like virus 1 (FoOuLV1) is a novel hypovirulence-inducing ourmia-like mycovirus. It belongs to the Botourmiaviridae family and can significantly reduce the pathogenicity of host fungi [27]. In this study, we identified mycoviruses in FoN isolated from the three major watermelon-production areas. Among them, most mycoviruses are identified from strains with an abnormal morphology. Therefore, we speculate that mycoviral infection is likely the main reason for the abnormal colony morphology of these strains. The FoN strains carrying hypovirulence mycoviruses has biocontrol potential for controlling *Fusarium* wilt of watermelon. In addition, we have also identified mycoviruses in normal strains, indicating that some viral infections do not alter the biological characteristics of fungal hosts. In this study, we found the phenomenon of co-infection in the tested strains through RT-PCR. Co-infection is a prevalent phenomenon where multiple viruses simultaneously infect the same fungus under natural conditions, and this has been observed in many fungal species [28]. For instance, it has been reported that a strain of *Sclerotinia sclerotiorum* was infected by nine different mycoviruses, which were classified into eight distinct viral families [29]. Similarly, co-infection has also been documented in *Purpureocillium lilacinum* and *Beauveria bassiana*, two significant entomopathogenic fungi, as well as in various other fungal pathogens [30,31]. However, the effect of mycoviral co-infection on FoN needs further study.

The identification of diverse mycoviruses in FoN, particularly the eight novel viruses, opens promising avenues for biocontrol of *Fusarium* wilt. At present, hypovirulence-associated mycoviruses could be deployed through two primary strategies: (1) the direct application of viral particles or infected fungal strains to soil or seedlings, or (2) horizontal transmission by engineering hypovirulent Fon isolates as “Trojan horse” vectors to spread viruses within pathogenic populations [32]. In southern humid area, the dominance of dsRNA Chrysoviridae, which is known to dysregulate fungal virulence through RNA interference pathways [33], supports their use as region-specific biocontrol agents. These viruses could be applied via soil drenching or hypovirulent FoN strains, with biochar amendments enhancing moisture retention to sustain viral activity. Conversely, in arid northern regions, +ssRNA viruses, like Mitoviridae, may require clay-based nanocarrier encapsulation to stabilize their viability under low humidity [34], coupled with drip irrigation systems to optimize viral dispersal. The cross-regional *Fusarium oxysporum* potyvirus (FoNPV1), exhibiting a pan-China distribution, holds promise as a broad-spectrum biocontrol agent, akin to the transcontinental success of Sclerotinia sclerotiorum hypovirulence-associated DNA virus 1 [35]. Such geographically tailored strategies, integrating viral consortia with local agronomic practices, could maximize field efficacy while addressing climatic constraints.

The application of mycoviruses as biocontrol agents against phytopathogenic fungi faces several critical challenges. First, host specificity limits their broad-spectrum utility, as most mycoviruses exhibit narrow host ranges, often restricted to specific fungal strains or species. Second, environmental instability-particularly susceptibility to UV radiation, temperature fluctuations, and desiccation-compromises viral viability in field conditions. Third, the risk of fungal resistance evolution necessitates the long-term monitoring of pathogen-virus coevolution dynamics. Additionally, regulatory frameworks for mycovirus deployment remain underdeveloped, with unresolved biosafety concerns regarding unintended ecological impacts on non-target microorganisms. Future research should focus on multi-omics to decode hypovirulence mechanisms, such as the viral suppression of host RNA silencing. Genetic engineering (e.g., CRISPR-mediated host range expansion) and synthetic chimeric viruses could address specificity limitations. Advanced formulations, like clay-polymer nanocomposites, may enhance environmental resilience. Field trials must prioritize integrated strategies, combining mycoviruses with biocontrol microbes or resistant cultivars. Regionally tailored consortia-blending climate-adapted viral types (e.g., humidity-tolerant dsRNA and drought-resistant +ssRNA viruses)-could optimize cross-environment performance. Crucially, global collaboration is needed to establish standardized biosafety protocols and policy frameworks, ensuring the safe scaling of mycovirus biocontrol. By merging virology, agronomy, and nanotechnology, mycoviruses may evolve from experimental tools to sustainable solutions for managing crop fungal diseases in a climate-changing world.

## 5. Conclusions

This study presents the first large-scale areal analysis of FoN-associated mycoviruses in China, revealing significant areal variations in the virome composition across three major watermelon-growing areas. By identifying 25 viral genome segments, including eight novel mycoviruses, our findings enrich the mycoviruses genetic database. The observed regional disparities-higher mycoviral diversity in humid southern areas and a predominance of +ssRNA viruses in arid areas-suggest that environmental factors, such as humidity and temperature, shape mycoviral diversity. Beyond its ecological significance, this study lays the groundwork for future research on harnessing mycoviruses for biocontrol. While mycovirus-mediated hypovirulence holds promise for managing *Fusarium* wilt in watermelon, key challenges remain. These include improving the stability and transmissibility of mycoviruses under field conditions, expanding the host range specificity to cover diverse *Fusarium* strains, and developing effective delivery methods for large-scale agricultural application. Future studies should explore strategies to enhance mycoviral persistence in field environments, engineer viral strains with optimized biocontrol efficacy, and integrate mycovirus-based approaches into holistic disease management frameworks. By linking mycoviral diversity to biocontrol potential, this research supports the development of sustainable disease-management strategies to safeguard watermelon production, a vital economic crop in China.

## Figures and Tables

**Figure 1 microorganisms-13-00906-f001:**
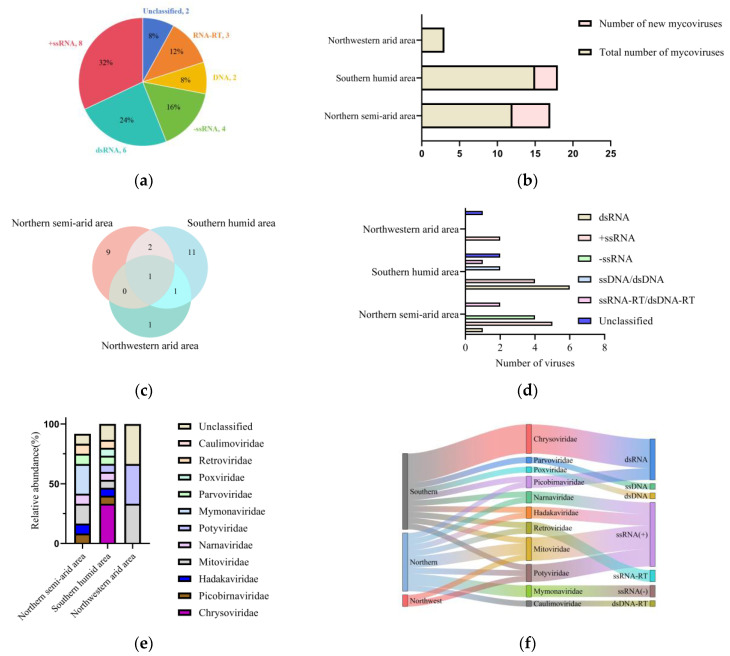
The diversity and the abundances of mycoviruses detected in FoN from different areas. (**a**) The percentage of each virus genome type. (**b**) The number of mycoviruses in three areas. (**c**) The number of shared mycoviruses in the virome of FoN in three areas. The numbers in different colors represent either the quantity of region-specific fungal viruses or the number of fungal viruses shared between areas. (**d**) Mycoviruses belonging to different genome types present in each area. (**e**) The relative abundance of mycoviruses at the family level in the same area, normalized by column. (**f**) The Sankey diagram shows the distribution of mycoviruses in three areas. Different colors represent distinct areas, taxonomic families, and gene types.

**Figure 2 microorganisms-13-00906-f002:**
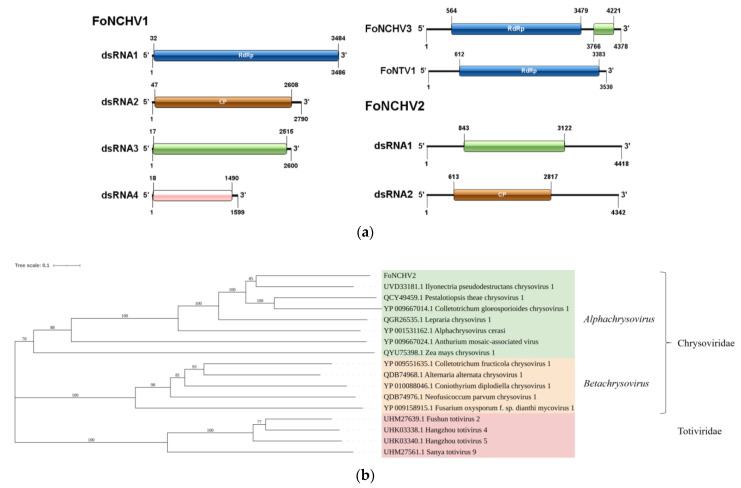
Gene structure of double-stranded RNA virus and phylogenetic relationships of FoNCHV2. (**a**) Gene structure of FoNCHV1, FoNCHV2, FoNCHV3, and FoNTV1. (**b**) NJ phylogenetic analysis depicting the evolutionary relationships of FoNCHV2.

**Figure 3 microorganisms-13-00906-f003:**
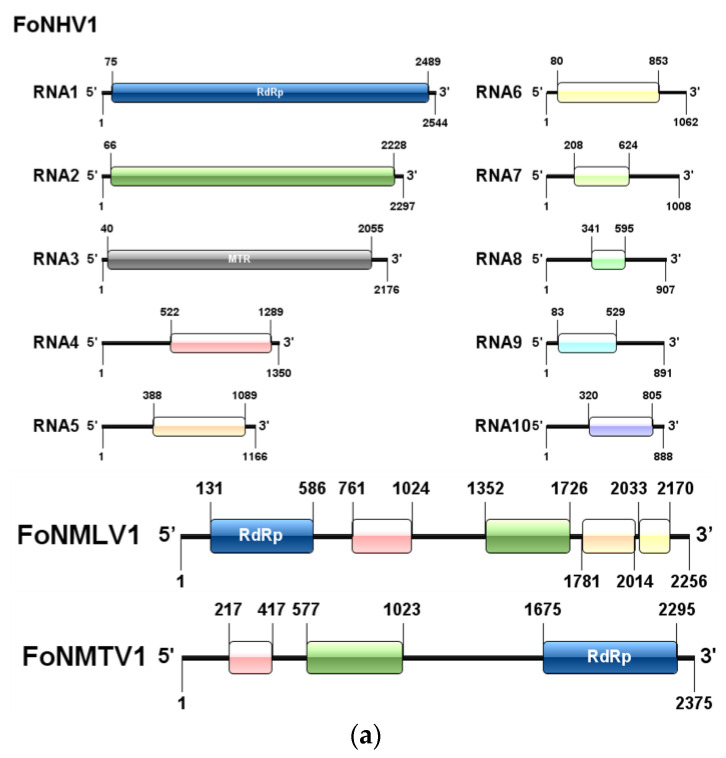
Gene structure of positive-sense single-stranded RNA viruses and phylogenetic relationships of FoNMTV1. (**a**) Gene structure of FoNHV1, FoNMTV1, and FoNTV1. (**b**) NJ phylogenetic analysis depicting the evolutionary relationships of FoNMTV1.

**Figure 4 microorganisms-13-00906-f004:**
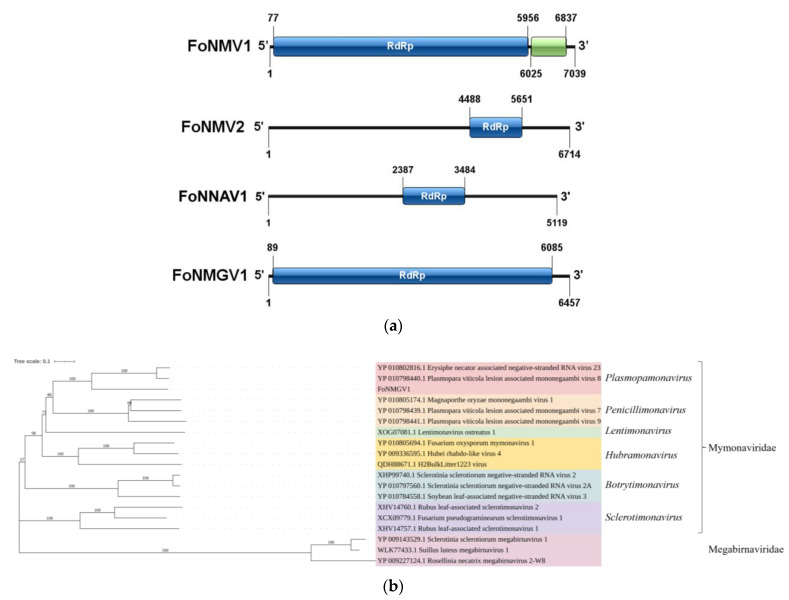
Gene structure of negative-sense single-stranded RNA viruses and phylogenetic relationships of FoNMGV1. (**a**) Gene structure of FoNMV1, FoNMV2, FoNNAV1, and FoNMGV1. (**b**) NJ phylogenetic analysis depicting the evolutionary relationships of FoNMGV1.

**Figure 5 microorganisms-13-00906-f005:**
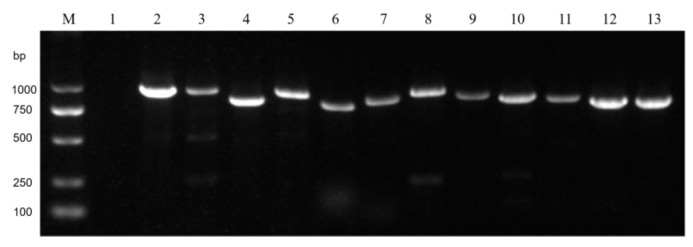
RT-PCR validation results. Electrophoresis in a 1.2% agarose gel. Lanes: M: DNA Marker 2000 bp. 1: Negative control, 2, 3: FoNNAV1, 4, 5: FoNMV1, 6, 7: FoNMTV1, 8, 9: FoNMLV1, 10, 11: FoNTV1, 12, 13: FoNCHV2.

## Data Availability

The data presented in this study are openly available in the NCBI Sequence Read Archive, with the accession number PRJNA1158555.

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
