# Peer review of "Mycoviral Diversity of *Fusarium oxysporum* f. sp. *niveum* in Three Major Watermelon-Production Areas in China"

_microorganisms, 2025, doi:10.3390/microorganisms13040906_

Round 1

Reviewer 1 Report

Comments and Suggestions for Authors

Manuscript: microorganisms-3543517

Title: Mycoviral diversity of Fusarium oxysporum f. sp. niveum in three major watermelon production areas in China

The manuscript delves into the diversity of mycoviruses within Fusarium oxysporum f. sp. niveum (Fon), a fungus causing watermelon wilt in China. The authors describe that through high-throughput sequencing on 150 Fon isolates from three primary watermelon cultivation areas, they identified 25 mycoviral sequences, including eight new mycoviruses. The authors claim these mycoviruses display varied distribution and diversity across different climatic regions, with significant implications for biocontrol strategies and watermelon disease management.

Overall comments:

The manuscript addresses a significant agricultural issue. The use of high-throughput sequencing to elucidate the mycovirus diversity in Fon is the best available tool and contributes valuable new insights into potential biocontrol methods.

The research aligns well with existing literature and introduces new findings that could enhance sustainable agricultural practices. However, the manuscript would benefit from a more detailed discussion on the practical applications of these findings in real-world agricultural settings. Additionally, it would be helpful to explore how these insights can be implemented to more effectively control Fusarium wilt.

Specific Comments, suggestions, and questions:

Abstract:

Line17 – 150 Fon isolated from (…). Do the authors mean 150 Fon stains?

Line 29 - Full virus name needs to be given the first time it is introduced.

Introduction:

Line 43 – “largest producer of watermelons.” In the world? If so then the sentence needs to be completed.

Line 54- The authors use Fon, FoN(before the virus name) and FON throughout the manuscript. This needs to be corrected and substituted by the correct notation throughout.

Lines 63-66 – Sentence needs revision.

Lines 68-70 – The meaning of this sentence is difficult to understand. Please revise for clarity.

Overall the introduction effectively sets the stage for the study but could be enhanced by briefly discussing previous efforts in biocontrol using mycoviruses, providing context for the significance of finding new mycoviral agents.

M&Ms:

The whole section needs revision as some of the subsections are missing and others seems out of order.

Line 115 – what is the purpose of this link?

Lines 127-132 – It is not clear if this culture step was done to obtain material for the HTS and/or for the morphological characterization of the different isolates. Either way, it needs to be clarified and the methodology concerning the statements in lines 397-399 need to be included, described and explained. It is not sufficient to claim that 18% of the strains had hypovirulence characteristics. Evidence needs to be provided and as it is it is missing.

Lines 134-140- The methodology here described is confusing. Lines 134-136 repeat what was already written in lines 131.132. So was this a different culture step? To what purpose?

Why 50 strains? How many stains from each watermelon growing region? What is meant by the same group (line 137)? Why were the mycelia combined?

Lines 144-145 – What are BNR and MNR groups? They are mentioned here and nowhere else in the manuscript. This is confusing and somewhat aggravating to the reader.

Lines 146-159 - Subsection 2.3 is supposed to describe library preparation and sequencing. But, instead, what it describes is the treatment of the raw data obtained after library preparation and sequencing. So an important part of the methodology is completely missing and cannot be evaluated.

Line 182-183 – what was the software used to design the primers, and which were the target regions?

Overall, the M&Ms section needs a thorough revision as it misses important information, and it is not clear on key points vital for the reader to follow the work done. Could the authors clarify if any controls were used during the sequencing to ensure no cross-contamination occurred among samples?

Results:

Line 190 – 601 samples – This is mentioned here for the first time in the manuscript. This type of information needs to be given in the M&Ms as it is part of the experimental design and the sampling effort.

Line 128 – 415 isolates – Same as in the previous point, this information only appears here in the results and no indication of how many isolates retrieved from each samples region is given.

Lines 195-196 – The reader does not know if the samples collected are representative or not because no information is given about how many samples collected at each region and how many strains/isolates obtained at each location. Also, it is not clear what is meant by sample and by isolate. Are these words being used as synonyms?

Lines 219-221- The areas did not detect the viruses. The authors did. Please correct the sentences.

Line 240- mycovirus detected in the northwestern arid area.

Line 249 – present

Line 251 – in the three areas.

Line 286-291 – The text is difficult to read. Please re-write for clarity.

Line 324 – Why was FonMV1 not included in the dendrogram in Fig. 4b?

Lines 328-329 – Why was QVU39969.1 not included in the dendrogram in Fig 4b?

Lines 338-348 – Why were the accessions OQ463847 (ThMV1) and OQ463848 (ThMV2) not included in the analysis?

Lines 357-365 – Are the M&Ms for this analysis the ones mentioned in section 2.5? If so, then the title of the section should be clarified and the number of analyzed Fon isolates from each area should be indicated.

Discussion:

 Line 370 – What is meant by regional differences?

Line 374 – Twelve

Line 376 – Re-write the sentence so it does not start with And.

Lines 383 and 391 – mycovirus-Fon – what does this mean?

Lines 397-399 – Where is the evidence for this claim?

Lines 408-429 – There is no analysis described in the manuscript to substantiate that in the present cases co-infection affects the pathogenicity of the Fusarium isolates. This part of the discussion needs to be shortened and toned down. The authors did not show that this effect was present in any of the isolates.

Lines 441-450 – This part of the discussion is purely speculative and unsubstantiated. In fact, there are no references cited here. It needs to be shortened or even taken out.

A discussion more focused about the potential for mycoviruses as biocontrol agents would seem more promisingand would strengthen this section. Could the authors speculate on how these mycoviruses might be applied in field conditions or integrated into existing disease management practices?

Conclusion:

Areas, Regional, Geographical are used in the manuscript seemingly as synonyms. However, the spatial scope that each of the terms convey is very different. Please ensure consistency in terminology and definitions throughout the manuscript to aid clarity.

Overall, the conclusion section is more of a summary. It needs to be revised to convey the main ideas as a take-home message for the reader. Consider expanding on the future research directions, particularly in overcoming challenges with mycovirus application in biocontrol.

These detailed points aim to refine the manuscript further, potentially increasing its impact and the clarity of its contributions to the field of plant pathology and biocontrol.

Author Response

Abstract:

Line17 – 150 Fon isolated from (…). Do the authors mean 150 Fon stains?

Response: The meaning of 150 Fon is 150 Fon strains, which we have revised it in the line 17.

Line 29 - Full virus name needs to be given the first time it is introduced.

Response:We have added the full virus name “Fusarium oxysporum f. sp. niveum Potyvirus 1” in lines 29-30 as your suggestion.

Introduction:

Line 43 – “largest producer of watermelons.” In the world? If so then the sentence needs to be completed.

Response: We have added“in the world”in the line 45.

Line 54- The authors use Fon, FoN(before the virus name) and FON throughout the manuscript. This needs to be corrected and substituted by the correct notation throughout.

Response: We have changed both FON and Fon to FoN throughout the article as your suggestion.

Lines 63-66 – Sentence needs revision.

Response:We have rewritten the sentence as your suggestion.

Lines 68-70 – The meaning of this sentence is difficult to understand. Please revise for clarity.

Response:We have rewritten the sentence as your suggestion.

Overall the introduction effectively sets the stage for the study but could be enhanced by briefly discussing previous efforts in biocontrol using mycoviruses, providing context for the significance of finding new mycoviral agents.

Response: we have revised the introduction as your suggestion. We have briefly discussed previous efforts in biocontrol using mycoviruses (References 15-18), providing context for the significance of finding new mycoviral agents in lines 98-102.

M&Ms:

The whole section needs revision as some of the subsections are missing and others seems out of order.

Line 115 – what is the purpose of this link?

Response:The climate information of the three production areas in the manuscript comes from this link.

Lines 127-132 – It is not clear if this culture step was done to obtain material for the HTS and/or for the morphological characterization of the different isolates. Either way, it needs to be clarified and the methodology concerning the statements in lines 397-399 need to be included, described and explained. It is not sufficient to claim that 18% of the strains had hypovirulence characteristics. Evidence needs to be provided and as it is it is missing.

Response:We have revised the Materials and Methods section to separately describe the culturing steps for strain morphology observation (lines 142-143) and HTS sequencing (lines 154-156). We have also included the methods for biological characterization (lines 144-152) and the corresponding results (Fig. S1 and Fig. S2). The statement in lines 397-399 that "18% of the strains exhibited hypovirulent traits" is inaccurate; it should be "18% of the strains showed abnormal biological characteristics." We have added morphological images, growth rate, and sporulation data for 27 strains (18% of the total strains) in Fig. S1 and Fig. S2.

Lines 134-140- The methodology here described is confusing. Lines 134-136 repeat what was already written in lines 131.132. So was this a different culture step? To what purpose?

Response:We have deleted the duplicate description.

Why 50 strains? How many stains from each watermelon growing region? What is meant by the same group (line 137)? Why were the mycelia combined?

Response:We isolated a total of 415 FoN strains from three watermelon-producing areas (northern, southern, and northwestern production area) with 140, 162, and 113 strains respectively. We have added this data in lines 139. Since the number of pathogenic isolates varied across the three areas, to ensure equal representation during sequencing, we randomly selected 50 strains from each region. The mycelia of 50 strains from same area were mixed together as a treatment for sequencing.

Lines 144-145 – What are BNR and MNR groups? They are mentioned here and nowhere else in the manuscript. This is confusing and somewhat aggravating to the reader.

Response:BNR and MNR groups represent two different biological duplication groups, which are not explained in the manuscript, which will make readers confused. We have deleted them in the manuscript.

Lines 146-159 - Subsection 2.3 is supposed to describe library preparation and sequencing. But, instead, what it describes is the treatment of the raw data obtained after library preparation and sequencing. So an important part of the methodology is completely missing and cannot be evaluated.

Response:We have added the methods of library preparation and sequencing, and Virus-related sequence assembly and annotation after raw data obtaining.

Line 182-183 – what was the software used to design the primers, and which were the target regions?

Response: We have added the primer design software to line 214, and the target region to Table S1.

Overall, the M&Ms section needs a thorough revision as it misses important information, and it is not clear on key points vital for the reader to follow the work done. Could the authors clarify if any controls were used during the sequencing to ensure no cross-contamination occurred among samples?

Response: We did not set up controls to prevent cross-contamination during sequencing. After reviewing the literature, we did not find any established methods for setting controls to prevent cross-contamination during sequencing, and the references we consulted also did not include such controls. To minimize the risk of cross-contamination, we used separate tools for each sample when collecting mycelia for sequencing, and all tools underwent strict sterilization and disinfection procedures to ensure no cross-contamination between samples..

  1. &nbspJia, J.; Fu, Y.; Jiang, D.; Mu, F.; Cheng, J.; Lin, Y.; Li, B.; Marzano, S.-Y.L.; Xie, J. Interannual Dynamics, Diversity and Evolution of the Virome in Sclerotinia Sclerotiorum from a Single Crop Field. Virus Evol2021, 7, doi:10.1093/ve/veab032.
  2. &nbspHe, Z.; Huang, X.; Fan, Y.; Yang, M.; Zhou, E. Metatranscriptomic Analysis Reveals Rich Mycoviral Diversity in Three Major Fungal Pathogens of Rice. Int J Mol Sci2022, 23, 9192, doi:10.3390/ijms23169192.
  3. &nbspWang, Q.; Cheng, S.; Xiao, X.; Cheng, J.; Fu, Y.; Chen, T.; Jiang, D.; Xie, J. Discovery of Two Mycoviruses by High-Throughput Sequencing and Assembly of Mycovirus-Derived Small Silencing RNAs From a Hypovirulent Strain of Sclerotinia Sclerotiorum. Front Microbiol2019, 10, doi:10.3389/fmicb.2019.01415.

Results:

Line 190 – 601 samples – This is mentioned here for the first time in the manuscript. This type of information needs to be given in the M&Ms as it is part of the experimental design and the sampling effort.

Response:We have added the information in lines136-137 of M&Ms as your suggestion.

Line 128 – 415 isolates – Same as in the previous point, this information only appears here in the results and no indication of how many isolates retrieved from each samples region is given.

Response:We have added the information in line 139 of M&Ms as your suggestion.

Lines 195-196 – The reader does not know if the samples collected are representative or not because no information is given about how many samples collected at each region and how many strains/isolates obtained at each location. Also, it is not clear what is meant by sample and by isolate. Are these words being used as synonyms?

Response: We have added information on the number of samples collected and the number of pathogen strains isolated in each region to 2.1 of M&Ms.

"sample" refers to the collected watermelon wilt-infected plant, while "strain" refers to the Fusarium oxysporum f. sp. niveum strain isolated from the infected plant.

Lines 219-221- The areas did not detect the viruses. The authors did. Please correct the sentences.

Response: We have revised the sentence in lines 249-250 as your suggestion.

Line 240- mycovirus detected in the northwestern arid area.

Response: We have revised it as your suggestion.

Line 249 – present

Response: We have revised it as your suggestion.

Line 251 – in the three areas.

Response: We have revised it as your suggestion.

Line 286-291 – The text is difficult to read. Please re-write for clarity.

Response: We have rewritten this section as your suggestion.

Line 324 – Why was FonMV1 not included in the dendrogram in Fig. 4b?

Response: In the method, we have explained that sequences of new viruses are used to build dendrogram to identify new virus species. The RdRp encoded by FoNMV1 shows the highest similarity (73.5%) to that of Plasmopara viticola lesion associated mymonavirus 1 (YP_010798438.1). According to the classification criteria of the ICTV (where an RdRp sequence similarity of less than 70% indicates a new species of genus Sclerotimonavirus), Therefore, it shows that FoNMV1 is not a new virus, so FoNMV1 was not included in the dendrogram.

Lines 328-329 – Why was QVU39969.1 not included in the dendrogram in Fig 4b? Lines 338-348 – Why were the accessions OQ463847 (ThMV1) and OQ463848 (ThMV2) not included in the analysis?

Response: By sequence alignment, very few sequences were found to be similar to the RdRp sequence of FoNMV2, with only three hits. Even the highest similarity, between the RdRp sequence of FoNMV2 and QVU39969.1, was only 29.1%. Due to the limited number of similar sequences and their low similarity, it was not possible to construct a phylogenetic tree to determine the classification of FoNMV2.

Similarly, very few sequences were found to be similar to the RdRp sequence of FoNNAV1, with only two hits. Although the highest similarity of FoNNAV1's RdRp sequence was 52.75% with that of Grapevine-associated negative single-stranded RNA virus 1, the classification of Grapevine-associated negative single-stranded RNA virus 1 remains unknown. Additionally, due to the limited number of similar sequences, constructing a phylogenetic tree to determine the classification of FoNNAV1 was not feasible.

This explanation has already been provided in lines 365-367.

Comparison results

Accession

Scientific Name

Percent Identity

FoNMV2

QVU39969.1

Magnaporthe oryzae mymonavirus 2

29.10%

QJW70352.1

Erysiphe necator associated negative-stranded RNA virus 21

26.51%

QXN75390.1

Grapevine-associated mononega-like virus 6

27.31%

FoNNAV1

QXN75409.1

Grapevine-associated negative single-stranded RNA virus 1

52.75%

QJW70365.1

Erysiphe necator associated negative-stranded RNA virus 17

45.06%

Lines 357-365 – Are the M&Ms for this analysis the ones mentioned in section 2.5? If so, then the title of the section should be clarified and the number of analyzed Fon isolates from each area should be indicated.

Response: We have added relevant information in lines219-221 of M&Ms.

Discussion:

 Line 370 – What is meant by regional differences?

Response:We have revised the original sentence to “we found that the diversity of mycoviruses in Fusarium oxysporum f. sp. Niveum differ significantly in different regions” in lines 404-405. 

Line 374 – Twelve

Response:We have changed 12 to twelve as your suggestion.

Line 376 – Re-write the sentence so it does not start with And.

Response:We have rewritten the sentence as your suggestion in lines 410-412.

Lines 383 and 391 – mycovirus-Fon – what does this mean?

Response: In order to understand this sentence more clearly, we have revised the original sentence to “providing an extended survival environment for FoN, the host of mycoviruses”, in lines 418 and 427.

Lines 397-399 – Where is the evidence for this claim?

Response: The statement that "18% of the strains exhibited hypovirulent traits" is inaccurate; it should be "18% of the strains showed abnormal biological characteristics." We have added morphological images, growth rate, and sporulation data for 27 strains (18% of the total strains) in Fig. S1 and Fig. S2.

Lines 408-429 – There is no analysis described in the manuscript to substantiate that in the present cases co-infection affects the pathogenicity of the Fusarium isolates. This part of the discussion needs to be shortened and toned down. The authors did not show that this effect was present in any of the isolates.

Response: We have shortened this part according to your suggestion.

Lines 441-450 – This part of the discussion is purely speculative and unsubstantiated. In fact, there are no references cited here. It needs to be shortened or even taken out.

Response: We have deleted this part of the discussion according to your suggestion.

A discussion more focused about the potential for mycoviruses as biocontrol agents would seem more promisingand would strengthen this section. Could the authors speculate on how these mycoviruses might be applied in field conditions or integrated into existing disease management practices?

Response: We have added 2 paragraphs of discussion according to your suggestion.

Conclusion:

Areas, Regional, Geographical are used in the manuscript seemingly as synonyms. However, the spatial scope that each of the terms convey is very different. Please ensure consistency in terminology and definitions throughout the manuscript to aid clarity.

Response: we have used areas uniformly throughout the article according to your suggestion.

Overall, the conclusion section is more of a summary. It needs to be revised to convey the main ideas as a take-home message for the reader. Consider expanding on the future research directions, particularly in overcoming challenges with mycovirus application in biocontrol.

Response: We have rewritten the conclusion according to your suggestion.

These detailed points aim to refine the manuscript further, potentially increasing its impact and the clarity of its contributions to the field of plant pathology and biocontrol.

Response: Thank you for your valuable feedback and constructive suggestions. We appreciate your insightful comments, which have helped us refine our manuscript and enhance its impact in the field of plant pathology and biocontrol.

Reviewer 2 Report

Comments and Suggestions for Authors

Review of manuscript: Mycoviral diversity of Fusarium oxysporum f. sp. niveum in three major watermelon production areas in China. This manuscript presents a study on the diversity of mycoviruses associated with Fusarium oxysporum f. sp. niveum (Fon) in three major watermelon-growing regions in China. This research is pertinent to the search for organic methods of plant disease control, especially in the face of limitations of traditional measures, such as pesticides and resistant varieties. The interpretation of the results is well grounded in the scientific literature, with the authors highlighting the potential of mycoviruses as biocontrol agents, a notion that aligns with previous studies on their ability to induce hypovirulence in pathogenic fungi. The discussion also considers the importance of geographical differences in mycovirus species composition for future biocontrol strategies. However, some aspects require refinement, such as a more detailed discussion on the potential mechanisms of hypovirulence of new mycoviruses. I think a more in-depth analysis of the ecological role of mycoviruses in different crop environments is also required. The methodology, which is generally described in detail and clearly, lacks a section on the laboratory preparation of libraries for sequencing and sequencing parameters, i.e., read length, depth, sequencer type, and model. It is as if the authors have accidentally omitted one section from the methodology. 

Comments on the Quality of English Language

The manuscript contains minor linguistic and stylistic errors, which can be readily corrected. Focusing on precision, conciseness, and clarity of language will undoubtedly enhance the quality of the text.

Author Response

Some aspects require refinement, such as a more detailed discussion on the potential mechanisms of hypovirulence of new mycoviruses. I think a more in-depth analysis of the ecological role of mycoviruses in different crop environments is also required. 

Response: we have add the discussion on the potential mechanisms, the field application method in different regions or climate conditions and the future research direction of hypovirulence of new mycoviruses in lines 457-497.

The methodology, which is generally described in detail and clearly, lacks a section on the laboratory preparation of libraries for sequencing and sequencing parameters, i.e., read length, depth, sequencer type, and model. It is as if the authors have accidentally omitted one section from the methodology. 

Response:We have added the methods of library preparation and sequencing, and Virus-related sequence assembly and annotation after raw data obtaining in lines 165-189.
